# Association of Food Allergy, Respiratory Allergy, and Skin Allergy with Attention Deficit/Hyperactivity Disorder among Children

**DOI:** 10.3390/nu14030474

**Published:** 2022-01-21

**Authors:** Guifeng Xu, Buyun Liu, Wenhan Yang, Linda G. Snetselaar, Mingwu Chen, Wei Bao, Lane Strathearn

**Affiliations:** 1Department of Pediatrics, The First Affiliated Hospital of USTC, Division of Life Sciences and Medicine, University of Science and Technology of China, Hefei 230001, China; chenmingwu01@163.com; 2Center for Disabilities and Development, University of Iowa Stead Family Children’s Hospital, Iowa City, IA 52242, USA; lane-strathearn@uiowa.edu; 3Division of Life Sciences and Medicine, University of Science and Technology of China, Hefei 230026, China; buyunliu@ustc.edu.cn (B.L.); wbao@ustc.edu.cn (W.B.); 4Department of Nutrition and Food Hygiene, School of Public Health, Guangdong Pharmaceutical University, Guangzhou 510006, China; wenhan-yang@gdpu.edu.cn; 5Department of Epidemiology, University of Iowa, Iowa City, IA 52242, USA; linda-snetselaar@uiowa.edu; 6Division of Developmental Behavioral Pediatrics, Stead Family Department of Pediatrics, University of Iowa Carver College of Medicine, Iowa City, IA 52242, USA

**Keywords:** food allergy, respiratory allergy, skin allergy, attention deficit/hyperactivity disorder

## Abstract

Background: Previous studies have predominately examined associations of respiratory allergy and skin allergy with ADHD, but little is known about the association between food allergy and ADHD. Methods: We included 192,573 children aged 4–17 years from the National Health Interview Survey (NHIS), a leading health survey in a nationally representative sample of the US population. Allergy conditions and ADHD were defined based on an affirmative response in the NHIS questionnaire. We used weighted logistic regression to estimate the odds ratio (OR) of ADHD. Results: Among the 192,573 children, 15,376 reported ADHD diagnosis. The prevalence of ADHD was higher among children with allergic conditions: 12.66% vs. 7.99% among children with and without food allergy; 12.16% vs. 7.63% among children with and without respiratory allergy; and 11.46% vs. 7.83% among children with and without skin allergy. After adjusting for covariates, the OR of ADHD was 1.72 (95% CI, 1.55–1.91) comparing children with and without food allergy, 1.50 (95% CI, 1.41–1.59) comparing children with and without respiratory allergy, and 1.65 (95% CI, 1.55–1.75) comparing children with and without skin allergy. The observed associations remained significant after mutual adjustment for other allergic conditions. Conclusions: In a nationally representative sample of US children, we found a significant association of common allergic conditions (food allergy, respiratory allergy, and skin allergy) with ADHD.

## 1. Introduction

Attention deficit/hyperactivity disorder (ADHD) is a childhood-onset neurodevelopmental disorder that is characterized by hyperactivity, impulsivity, and/or inattention [1]. Epidemiologic data have shown a steady increase in the prevalence of ADHD [2,3]. In the United States, ADHD affects about 10% of children aged 4–17 years [3,4]. ADHD may be accompanied by emotional and behavioral problems in children, leading to a considerable financial burden on society (between USD 143 billion and USD 266 billion in incremental costs annually in the U.S.) [5]. Although genetic factors appear to play an important role in its etiology [6], a 67% increase in ADHD prevalence among US children over the past two decades (from 6.1% in 1997–1998 to 10.2% in 2015–2016) [3] suggests that environmental factors may also contribute to the etiology of this disorder.

Accumulating evidence supports the hypothesis that ADHD may be an inflammation and immune-associated disease [7] and suggests a potential link between allergy and ADHD [8]. Allergic conditions, including respiratory allergy, skin allergy, and food allergy, are common medical conditions in children [9]. In parallel with the increasing trend of ADHD [3], the prevalence of food and skin allergies increased steadily in annual surveys conducted from 1997 to 2011, despite no significant change in the prevalence of respiratory allergy [10]. Biologically, excessive release of cytokines and keratinocytes under allergic conditions may cause structural and functional changes in specific brain areas and ADHD behavioral patterns, therefore linking allergy to ADHD [11]. Additionally, peripheral immune cells could permeate across the blood–brain barrier [12], which could also affect the functions of neurons. In addition, some food allergens may interfere with intestinal microbiota [13,14] and affect the expression of the neurotransmitter serotonin (5-HT) [15], which plays an important role in ADHD etiology [16].

Previous studies examining the association between allergic conditions and ADHD are predominately focused on respiratory allergy and skin allergy [17,18]. For example, a meta-analysis including 25 studies about asthma and ADHD found a significant association between asthma and ADHD, with an odds ratio of 1.52 [19]. Hospital-diagnosed atopic dermatitis was found to be associated with ADHD in a case-control study in Danish children [20]. Little is known regarding the association between food allergy and ADHD [21]. Moreover, the findings in previous studies are inconsistent. For example, some studies found a significant association of allergic rhinitis, atopic dermatitis, and asthma with ADHD [22], while other studies reported no significant association between ADHD and allergic disorders [23]. The inconsistent findings may be at least partly due to the limited sample size and statistical power in some of the previous studies. 

Therefore, in this study, we analyzed large-scale and nationally representative data to examine the associations of food allergy and other allergic conditions with ADHD in US children. We hypothesized that children with food allergy may have a higher risk for ADHD than children without food allergy.

## 2. Methods

### 2.1. Study Population

The National Health Interview Survey (NHIS) is a leading health survey conducted annually by the National Center for Health Statistics at the Centers for Disease Control and Prevention. The NHIS, with a nationally representative sampling, collects comprehensive and detailed data on a broad range of health topics from the US population. Since 1957, it has become the principal source of information on the health conditions of the US population [24]. The annual sample size of the NHIS is about 35,000 households containing about 87,500 persons. The study design and methodology of the NHIS were published elsewhere [25,26]. 

### 2.2. Ascertainment of Variables

The NHIS conducts in-person household interviews to collect data for all household members, including children. For each interviewed family in the household, one sample child (if any children aged ≤ 17 years are present) is randomly selected by a computer program, and no differential sampling probabilities are applied to the children [26]. Detailed health-related information, including information on physical and mental health, is collected for the sample child. This information is provided by an adult, usually a parent, who is knowledgeable about the child’s health.

Allergic conditions were defined based on an affirmative response to the following separate questions [27]: “During the past 12 months, has [your child] had (1) any kind of food or digestive allergy; (2) any kind of respiratory allergy; (3) eczema or any kind of skin allergy?”. ADHD was defined based on an affirmative response to the question [3]: “Has a doctor or health professional ever told you that [your child] had Attention Deficit Hyperactivity Disorder (ADHD) or Attention Deficit Disorder (ADD)?”.

Demographic data, including age, sex, race/ethnicity, education, family income, and geographic region, were collected using a standardized questionnaire during the interview. Race and Hispanic ethnicity were self-reported and classified based on the 1997 Office of Management and Budget Standards. Family income levels were classified according to the ratio of family income to federal poverty level (<1.0, 1.0–1.9, 2.0–3.9, and ≥4.0).

### 2.3. Statistical Analysis

We used survey sampling weights, strata, and primary sampling units created by the NCHS and provided along with the NHIS data in all the analyses, unless otherwise specified, so that the results are nationally representative of the US population. 

Comparisons of baseline characteristics among children with and without food allergy or other allergic conditions were performed using linear regression for continuous variables and the chi-square test for categorical variables. We estimated the odds ratio (OR) and 95% confidence interval (CI) of ADHD according to the presence of allergic conditions using multivariable logistic regression, adjusting for age, sex, race/ethnicity, family highest education level, family income to poverty ratio, and geographic region. Because children with food allergy are more likely to have asthma and other allergies compared with children without food allergies [28], we further considered a mutual adjustment for other allergic conditions. 

To assess whether the association differs by population characteristics, we performed subgroup analyses according to age (4–11 or 12–17 years), sex (male or female), and race/ethnicity (white or non-white). Interactions between these factors and allergic conditions were tested by adding their multiplicative interaction terms in the multivariable models.

All data analyses were conducted using the survey procedures of SAS 9.4 (SAS Institute Inc., Cary, NC, USA). Two-sided *p* < 0.05 was considered statistically significant.

## 3. Results

Among the 192,573 children aged 4–17 years old included in this analysis, 8603 had food allergy, 24,218 had respiratory allergy, and 18,703 had skin allergy. Compared with children without food allergy, children with food allergy were more likely to be white, and they had higher family education levels (Table 1). Children with respiratory allergy, compared with those without respiratory allergy, were older, more likely to be male, white, and they had higher family education levels (Table 2). Children with skin allergy were younger, more likely to be black, and they had higher family education levels than children without skin allergy (Table 3).

A diagnostic history of ADHD was reported in 15,376 children. Children with ADHD were older, more likely to be male and white, and they had lower family income levels (Appendix A). The weighted prevalence of ADHD was higher among children with allergic conditions: 12.66% (95% CI 11.57–13.75) vs. 7.99% (95% CI 7.82–8.15) among children with and without food allergy (*p* < 0.001); 12.16% (95% CI 11.60–12.72) vs. 7.63% (95% CI 7.46–7.80) among children with and without respiratory allergy (*p* < 0.001); and 11.46% (95% CI 10.90–12.01) vs. 7.83% (95% CI 7.66–8.01) among children with and without skin allergy (*p* < 0.001). After adjustment for age, sex, race/ethnicity, family highest education level, family income level, and geographical region, the OR of ADHD was 1.72 (95% CI, 1.55–1.91) for food allergy, 1.50 (95% CI, 1.41–1.59) for respiratory allergy, and 1.65 (95% CI, 1.55–1.75) for skin allergy when comparing children with these conditions and those without. The observed associations were modestly attenuated but remained significant after mutual adjustment for other allergic conditions, with the corresponding ORs of 1.44 (95% CI, 1.29–1.60), 1.37 (95% CI, 1.28–1.45), and 1.49 (95% CI, 1.39–1.59), respectively (Table 4).

We observed a significant association between any kind of allergy and ADHD in all subgroups stratified by age, sex, and race/ethnicity (Table 5). There was a significant interaction between race/ethnicity and each allergic condition, with a stronger association in non-white children than white children. In addition, there was a significant interaction between sex and food or respiratory allergy, with a stronger association in female than in male. Sensitivity analyses by restricting to children whose information was reported by their parents rather than other household members (*n* = 166,255) yielded similar results (Appendix A).

## 4. Discussion

In a nationally representative sample of US children, we found a significant and positive association of food allergy, respiratory allergy, and skin allergy with ADHD. This association persisted after adjustment for demographic and socioeconomic variables, as well as each of the other types of allergic conditions. In addition, the association between each allergic condition and ADHD was significant in all subgroup analyses by age, sex, and race/ethnicity.

Our results have extended findings from earlier studies regarding allergic conditions, particularly food allergy and ADHD. Among the common allergic conditions (i.e., food allergy, skin allergy, and respiratory allergy), the association of respiratory allergy (e.g., asthma [22,29,30] and allergic rhinitis [31]) with ADHD has been most extensively examined in previous studies. A recent systematic review and meta-analysis showed a significant association between asthma and ADHD; the pooled adjusted OR was 1.53 (95% CI, 1.41–1.65) [22]. In a large population-based study with individuals in multiple national registers in Sweden, asthma was also significantly associated with ADHD (adjusted OR 1.45; 95% CI, 1.41–1.48) [22]. For skin allergy, a significant association was also reported in some, although not all, previous studies regarding the association of atopic dermatitis [24,32] and eczema [33] with ADHD. Previous studies concerning the association of food allergy with ADHD were sparse. A recent study among school-age children of 5–12 years old reported a significant association of early food allergy with ADHD [34]. In another study among children aged between 3 and 6 years, food allergy was also related to ADHD, but the association did not reach statistical significance [35]. 

In this study, all three common allergic conditions were significantly associated with ADHD, indicating that there might be shared mechanisms linking these allergic conditions to ADHD. Although the underlying mechanisms remain unclear, several direct and indirect pathways may be implicated in the link between allergic conditions and ADHD [36]. First, excessive release of cytokines and keratinocytes under allergic conditions may access the brain and affect neuronal activity of the anterior cingulate cortex (ACC) and prefrontal cortex (PFC) [37,38]. The structural and functional changes in PFC and ACC have been linked with deficits in attentional control, decision making, memory, and motor output, which are also considered main symptoms of ADHD [39]. Second, allergic disorders pose a unique stressor for affected children. Chronic stress to children may result from parental anxiety and overprotection to allergic disorders, as well as stigmatization and bullying. Chronic stress can cause stress sensitization, which may involve dysregulation of the hypothalamic–pituitary–adrenal (HPA) axis [40]. Abnormal cortisol reactivity due to HPA axis dysregulation can affect children’s executive functioning and attention [41]. Additionally, repeated stress has been shown to be associated with hippocampal, amygdala, and medial prefrontal cortex atrophy and dysfunction [42]. Differences in hippocampal and amygdala morphology have also been noted in ADHD patients [43]. Third, alterations in the gut–brain–behavior axis may be another link between food allergy and ADHD. Food-induced microbiome changes and allergic immune activation are thought to affect brain function through neuroimmune interactions, which can affect the enteric nervous system and central nervous system and eventually lead to neurodevelopmental abnormalities [44]. Fourth, inflammatory cytokines produced in allergic conditions could lead to altered metabolism of norepinephrine and dopamine, which was considered a critical neurological change in ADHD pathology [36,45]. Lastly, it is also possible that there is a shared cause leading to both allergic conditions and ADHD or a bidirectional association between allergic conditions and ADHD.

Special diets, especially elimination diets that are used to diagnose and treat food allergy, have been proposed as a therapeutic dietary approach for neurodevelopmental disorders including ADHD and autism spectrum disorder [46,47]. Several previous studies have examined the effects of elimination diets such as food additives exclusion diets and the oligoantigenic diets in ADHD [47]. The food additives exclusion diets attempt to exclude artificial food coloring, artificial flavors, artificial fragrances, preservatives, and artificial sweeteners, whereas the oligoantigenic diet attempts to exclude antigenic foods, such as cow’s milk, cheese, egg, chocolate, and nuts [47]. The effectiveness of food additives exclusion diets and the oligoantigenic diets in ADHD remains inconclusive. For gluten-free and/or casein-free diets, although such diets have been mainly investigated in autism spectrum disorder [48], little evidence is available to support or dispute their use in ADHD [47]. A number of previous studies have examined the association of Celiac disease, an autoimmune disease in which the intestine is hypersensitive to gluten, with ADHD, but their findings have been inconsistent [49,50,51].

A major strength of this study is the use of data from a nationally representative, multi-racial/ethnic population with a large sample size, which allows generalizing the findings to a broader population. In addition, the NHIS has a relatively high response rate, which further reduces the concern of selection bias. Several limitations of this study merit further consideration. First, there is no information about when allergies first happened and when ADHD diagnosis was first made; therefore, we could not establish a temporal relationship and causal inference from the current study. Future investigation in a large longitudinal cohort is warranted to confirm our findings. Second, the NHIS does not have laboratory data on specific IgE antibodies for the allergic conditions, and therefore, we were unable to attribute the observed associations to specific allergens. Third, as in previous research, possible differences in the association have not been explored across different types of ADHD (i.e., inattentive, hyperactive, combined), because information about ADHD subtypes is not available. This could be considered in future studies. 

## 5. Conclusions

The current study found a significant and positive association between common allergic conditions, including food allergy, respiratory allergy, and skin allergy, and ADHD in children. Although the detailed mechanisms linking food allergy and other allergic conditions to ADHD remain to be understood, physicians should be aware of the increased risk of ADHD as a comorbidity of children with allergic conditions.

## Figures and Tables

**Table 1 nutrients-14-00474-t001:** Characteristics of the participants (*n* = 192,573), according to food allergy status.

Variables	Children without Food Allergy	Children with Food Allergy	*p*-Value
No. of participants	183,970	8603	
Age, year	10.51 (0.01)	10.49 (0.06)	0.69
Sex			0.59
Male	94,808 (51.07%)	4345 (50.68%)	
Female	89,162 (48.93%)	4258 (49.32%)	
Race/ethnicity			<0.001
Hispanic	50,932 (20.17%)	1848 (16.23%)	
Non-Hispanic White	90,202 (58.05%)	4473 (59.46%)	
Non-Hispanic Black	28,138 (14.27%)	1345 (14.56%)	
Other	14,698 (7.51%)	937 (9.75%)	
Family highest education level			<0.001
Less than high school	39,929 (19.06%)	1263 (12.79%)	
High school	24,481 (13.22%)	945 (10.67%)	
College or higher	118,537 (67.22%)	6381 (76.41%)	
Missing	1023 (0.51%)	14 (0.14%)	
Family income to poverty ratio			<0.001
<1.0	26,506 (14.47%)	1247 (14.46%)	
1.0–1.9	33,834 (18.16%)	1541 (18.31%)	
2.0–3.9	46,761 (25.85%)	2199 (25.38%)	
≧4.0	40,539 (22.29%)	2287 (26.42%)	
Missing	36,330 (19.24%)	1329 (15.43%)	
Geographic region			<0.001
Northeast	30,638 (17.33%)	1713 (20.23%)	
Midwest	37,860 (23.66%)	1640 (21.74%)	
South	66,934 (36.28%)	2963 (34.59%)	
West	48,538 (22.74%)	2287 (23.44%)	

Data are presented as weighted means and standard errors in parentheses for continuous variables, and frequencies and weighted percentages in parentheses for categorical variables.

**Table 2 nutrients-14-00474-t002:** Characteristics of the participants (*n* = 192,573), according to respiratory allergy.

Variables	Children without Respiratory Allergy	Children with Respiratory Allergy	*p*-Value
No. of participants	168,355	24,218	
Age, year	10.48 (0.01)	10.74 (0.03)	<0.001
Sex			<0.001
Male	85,734 (50.38%)	13,419 (55.84%)	
Female	82,621 (49.62%)	10,799 (44.16%)	
Race/ethnicity			<0.001
Hispanic	47,919 (20.79%)	4861 (14.37%)	
Non-Hispanic White	80,833 (57.08%)	13,842 (65.43%)	
Non-Hispanic Black	25,860 (14.46%)	3623 (13.04%)	
Other	13,743 (7.67%)	1892 (7.16%)	
Family highest education level			<0.001
Less than high school	37,315 (19.49%)	3877 (13.76%)	
High school	22,761 (13.44%)	2665 (10.71%)	
College or higher	107,297 (66.54%)	17,621 (75.33%)	
Missing	982 (0.53%)	55 (0.19%)	
Family income to poverty ratio			<0.001
<1.0	24,536 (14.59%)	3217 (13.61%)	
1.0–1.9	31,291 (18.36%)	4084 (16.79%)	
2.0–3.9	42,353 (25.60%)	6607 (27.41%)	
≧4.0	36,330 (21.84%)	6496 (26.97%)	
Missing	33,845 (19.61%)	3814 (15.22%)	
Geographic region			0.003
Northeast	28,535 (17.61%)	3816 (16.42%)	
Midwest	34,644 (23.70%)	4856 (22.67%)	
South	59,426 (35.27%)	10,471 (42.82%)	
West	45,750 (23.43%)	5075 (18.09%)	

Data are presented as weighted means and standard errors in parentheses for continuous variables, and frequencies and weighted percentages in parentheses for categorical variables.

**Table 3 nutrients-14-00474-t003:** Characteristics of the participants (*n* = 192,573) by skin allergy status.

Variables	Children without Skin Allergy	Children with Skin Allergy	*p*-Value
No. of participants	173,870	18,703	
Age, year	10.56 (0.01)	10.06 (0.04)	<0.001
Sex			<0.001
Male	90,125 (51.33%)	9028 (48.61%)	
Female	83,745 (48.67%)	9675 (51.39%)	
Race/ethnicity			<0.001
Hispanic	48,756 (20.39%)	4024 (16.40%)	
Non-Hispanic White	85,609 (58.35%)	9066 (55.99%)	
Non-Hispanic Black	25,685 (13.79%)	3798 (18.75%)	
Other	13,820 (7.47%)	1815 (8.86%)	
Family highest education level			<0.001
Less than high school	38,311 (19.38%)	2881 (13.37%)	
High school	23,077 (13.18%)	2349 (12.38%)	
College or higher	111,476 (66.91%)	13,442 (74.08%)	
Missing	1006 (0.53%)	31 (0.16%)	
Family income to poverty ratio			<0.001
<1.0	24,859 (14.31%)	2894 (15.84%)	
1.0–1.9	31,845 (18.12%)	3530 (18.55%)	
2.0–3.9	43,969 (25.70%)	4991 (27.00%)	
≧4.0	38,244 (22.26%)	4582 (24.40%)	
Missing	34,953 (19.61%)	2706 (14.20%)	
Geographic region			0.48
Northeast	29,128 (17.52%)	3223 (16.97%)	
Midwest	35,580 (23.54%)	3920 (23.88%)	
South	63,190 (36.22%)	6707 (36.00%)	
West	45,972 (22.72%)	4853 (23.15%)	

Data are presented as weighted means and standard errors in parentheses for continuous variables, and frequencies and weighted percentages in parentheses for categorical variables.

**Table 4 nutrients-14-00474-t004:** Association of allergic conditions with ADHD (*n* = 192,573).

	Children without Specific Allergic Conditions	Children with Specific Allergic Conditions	*p*-Value
Food allergy			
No. of ADHD cases/total participants	14,338/183,970	1038/8603	
Model 1 ^a^	1.00 (reference)	1.72 (1.55–1.91)	<0.001
Model 2 ^b^	1.00 (reference)	1.72 (1.55–1.91)	<0.001
Model 3 ^c^	1.00 (reference)	1.44 (1.29–1.60)	<0.001
Respiratory allergy			
No. of ADHD cases/total participants	12,538/168,355	2838/24,218	
Model 1 ^a^	1.00 (reference)	1.59 (1.50–1.69)	<0.001
Model 2 ^b^	1.00 (reference)	1.50 (1.41–1.59)	<0.001
Model 3 ^c^	1.00 (reference)	1.37 (1.28–1.45)	<0.001
Skin allergy			
No. of ADHD cases/total participants	13,277/173,870	2099/18,703	
Model 1 ^a^	1.00 (reference)	1.67 (1.57–1.78)	<0.001
Model 2 ^b^	1.00 (reference)	1.65 (1.55–1.75)	<0.001
Model 3 ^c^	1.00 (reference)	1.49 (1.39–1.59)	<0.001

ADHD, attention deficit/hyperactivity disorder. ^a^ Model 1: adjusted for age and sex. ^b^ Model 2: model 1 plus race/ethnicity, family highest education level, family income to poverty ratio, and geographic region. ^c^ Model 3: model 2 plus mutual adjustment for other allergic conditions as mentioned.

**Table 5 nutrients-14-00474-t005:** Stratified analyses by age, sex, and race/ethnicity for the association of allergic conditions with ADHD.

		Food Allergy			Respiratory Allergy			Skin Allergy		
Variables	No. with ADHD/Total Participants	OR (95% CI) *	*p*-Value	*p* for Interaction	OR (95% CI) *	*p*-Value	*p* for Interaction	OR (95% CI) *	*p*-Value	*p* for Interaction
Age										
2–11 years	6695/104,482	1.48 (1.28–81.71)	<0.001	0.94	1.38 (1.26–1.52)	<0.001	0.81	1.52 (1.38–1.67)	<0.001	0.96
12–17 years	8681/8,8091	1.44 (1.23–1.68)	<0.001		1.32 (1.22–1.44)	<0.001		1.45 (1.32–1.59)	<0.001	
Sex										
Male	11,011/99,153	1.34 (1.17–1.53)	<0.001	0.02	1.30 (1.21–1.40)	<0.001	0.006	1.51 (1.39–1.64)	<0.001	0.86
Female	4365/93,420	1.64 (1.38–1.94)	<0.001		1.52 (1.36–1.70)	<0.001		1.44 (1.29–1.62)	<0.001	
Race/ethnicity										
White	9425/94,675	1.39 (1.21 (1.60)	<0.001	0.02	1.28 (1.19–1.37)	<0.001	<0.001	1.34 (1.23–1.47)	<0.001	<0.001
Non-white	5951/97,898	1.50 (1.28–1.76)	<0.001		1.58 (1.43–1.75)	<0.001		1.71 (1.54–1.90)	<0.001	

Abbreviations: CI, confidence intervals; OR, odds ratio. * Multivariable model adjusted for age, sex, race/ethnicity, family highest education level, family income to poverty ratio, geographic region, and mutual adjustment for other allergic conditions as mentioned, except the stratifying factor.

## Data Availability

The NHIS data can be accessed from the following website (https://www.cdc.gov/nchs/nhis/data-questionnaires-documentation.htm, accessed on 13 November 2021).

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
