# Peer review of "Association of Food Allergy, Respiratory Allergy, and Skin Allergy with Attention Deficit/Hyperactivity Disorder among Children"

_nutrients, 2022, doi:10.3390/nu14030474_

Round 1

Reviewer 1 Report

This is a very interesting study about the association of different types of allergies in children with attention deficit/hyperactivity disorder based on The National Health Interview Survey data . The manuscriprt is well written , with clear hypothesis and conclusions. The strength of the study is representative , large sample size. This type of the study does not allow to conclude on temporal relationship and causation , but the authors discuss it in the paper. The results are dispayed in a clear manner.

Author Response

We thank the reviewer for the positive comments. 

Reviewer 2 Report

I read the article with great interest, because food allergy is in the area of my scientific interests. The authors performed a valuable analysis of the correlation between ADHD and the occurrence of food, respiratory, and skin allergy in children. An undoubted advantage is the analysis based on the data of 192,573 children (aged 4-17 years).

However, I think the introduction and discussion are too brief. The correlation between allergy and neurological disorders is very extensive. Information on the role of the serotonin and opioid systems in children should be added, especially after receptor stimulation by opioid peptides (exo- and endogenous). It is also necessary to describe the aspect of the possibility of using a gluten-free and casein-free diet in the treatment of children with ADHD (in a similar way as in children with autism spectrum disorders).

Author Response

We thank the reviewer for the comments. We have reviewed the literature, and addressed those in the revised manuscript (page #6 and #7). 
